# How Managed Entry Agreements Influence the Patients’ Affordability to Biological Medicines—Bulgarian Example

**DOI:** 10.3390/healthcare11172427

**Published:** 2023-08-30

**Authors:** Zornitsa Mitkova, Ivan Manev, Konstantin Tachkov, Vladimira Boyadzhieva, Nikolay Stoilov, Miglena Doneva, Guenka Petrova

**Affiliations:** 1Faculty of Pharmacy, Medical University of Sofia, 1000 Sofia, Bulgaria; ivan.maneff@gmail.com (I.M.); ktashkov@pharmfac.mu-sofia.bg (K.T.); miglena_doneva@abv.bg (M.D.); guenka.petrova@gmail.com (G.P.); 2Rheumatology Clinic, University Hospital St. Ivan Rilski, Faculty of Medicine, Medical University of Sofia, 1612 Sofia, Bulgaria; vladimira.boyadzhieva@gmail.com (V.B.); dr_nstoilov@yahoo.com (N.S.)

**Keywords:** biologics, affordability, reimbursement, managed entry agreement

## Abstract

Managed entry agreements are applied in almost all European countries in order to improve patients’ access to therapy. The current study aims to evaluate the changes in the affordability of biological medicines for patients in Bulgaria during 2019–2022. The study is a top-down macroeconomic analysis of the key economic indicators and reimbursed costs of biologic therapies. Affordability was determined as the number of working hours needed to pay for monthly therapy. The average NHIF budget for pharmaceuticals increased significantly along with inflation in the healthcare sector. Bulgarian patients had to devote a large part of their income to buying medicines if a co-payment existed. The percentage of the monthly income of pensioners needed for therapy co-payment varied between 10% and 280%. The hours of work required to purchase a package of biologicals varied between 7 and 137 working hours. The global economic crisis has affected Bulgaria and led to worsening economic parameters. There are still no well-established practices to control public spending, as the measures taken to reduce the final cost of medicines mainly affect the pharmaceutical companies. This type of cost-containment policy provides an opportunity for innovative treatment with biologicals for patients with inflammatory diseases. Most of the therapies cost more than the patients’ monthly income.

## 1. Introduction

Biological therapy of patients with inflammatory diseases is associated with better outcomes and significant improvement in patients’ symptoms and quality of life [1]. High medicine prices are considered one of the main reasons for different patient access in countries because they limit the health care budget. Furthermore, different levels of patient co-payments have been established between European countries, making the treatment even more costly from the patients’ point of view [2,3].

Biosimilars might lower acquisition cost [1], thus reducing the healthcare expenditure and leading to cost savings. Switching from a biologic to a biosimilar is safe and effective, but additional follow-up of patients’ outcomes is needed [4]. Not all biologicals possess biosimilar alternatives, and therefore, other measures to reduce the costs of biological therapy are needed.

Managed entry agreements are different measures aiming to improve patients’ access to therapy. They include confidential discounts, paybacks, price–volume agreements, free doses, bundles and other agreements, payment by result, etc. [5]. Managed entry agreements are applied in almost all European countries, and their effect is under constant study [6,7].

A comparison of medicine prices and affordability could be used to illustrate public and private expenditure and the effectiveness of national policies towards improving access to biologicals and biosimilars. The affordability of therapy is related not only to the cost of therapy but also to the patient’s financial capacity to meet other basic needs after medicine payment. Measuring affordability includes an analysis of the prices of medicines and co-payments as well as personal income [8,9,10].

Measuring affordability allows both international comparisons and national public and individual payment analysis. This is a basic methodology that indicates whether medicines are affordable for patients and therefore whether the prescribed therapy can be purchased. Most of the studies are focused on calculating affordability for chronic disease and assessing patients’ ability to co-pay for monthly therapy [11,12].

Measuring the economic effect of managed entry agreements on patients’ access to medicines is also crucial for evaluating their effectiveness and further regulatory changes.

This study aims to evaluate the changes in the affordability of biological medicines for patients in Bulgaria during 2019–2022 within the general economic framework in the country. The study explores two groups of patients (retired and of working age) with a chronic inflammatory disease and on biological therapy who require long-term therapy.

## 2. Materials and Methods

### 2.1. Design of the Study

The study is a top-down macroeconomic analysis of the key economic indicators and reimbursed costs of biologic therapies during the period of 2019–2022. The main economic indicators were systematized by year and tendencies in their changes in percentages, and the average ± standard deviation (SD) was analyzed to outline the general economic framework.

The second stage of the study included a regulatory analysis of the legislation on reimbursement conditions and managed entry agreements for biologics in force during the period of 2019–2022.

As a third step, the affordability for patients of biologic therapy for inflammatory diseases was calculated by considering the monthly income of the population.

### 2.2. Affordability Calculation

Affordability was accessed via calculating patient income needed to purchase the monthly therapy. The World Health Organization (WHO) defines affordability as the number of working hours per month needed for patients to purchase medicines for monthly therapy [13]. Additionally, the percentage of pensioner income needed for monthly therapy was calculated. 

We first calculated the cost of therapy with 12 INNs available on the national market belonging to ATC codes L04AA29, L04AA37, L04AB01, L04AB02, L04AB04, L04AB05, L04AC05, L04AC06, L04AC07, L04AC10, L04AC13, and L01XC02. The cost was calculated by multiplying the price per dose unit and the recommended dosage regimen for annual treatment according to summary of the product characteristics.

The patient co-payment was established as the difference between the final medicine price and the value paid by the NHIF (25% co-payment). The sum was divided by 12 in order to find the patient co-payment for monthly therapy. The lowest and highest prices of medicines in the Positive Drug List (PDL) were extracted to calculate the monthly cost per patient.

Next, the average monthly wage was divided by the number of working hours per month to estimate the average wage per working hour. The affordability was determined as the number of working hours needed to pay for monthly therapy in the case of working patients and the percentage of total income necessary for monthly therapy in the case of pensioners.
(1)Wage per hour = [Average monthly wage]/[(22 working days) × (8 working hours)](2)Hourly wage needed for monthly therapy = (Medicinal product cost of therapy – part reimbursed by NHIF)/Average wage per hour


### 2.3. Data Sources

The following sources were used:

National Statistical Institute—officially published reports for economic indicators: average inflation; healthcare sector, medicinal products, outpatient visits, and hospital service inflation; difference in average wages; GDP difference (%) [14,15];

National Insurance Institute—average monthly pension (all types of pensions) [16];

National Council of pricing and reimbursement of medicinal products—Positive Drug List—biological medicine prices and patient co-payments during 2019–2022 [17];

National Health Insurance Fund—reimbursed cost of biological products (NHIF budget; reimbursed expenditures for biological products) [18].

### 2.4. Statistical Analysis

The range, mean value, and standard deviation (SD) of economic indicators were calculated with Excel 10 in order to compare the investigated parameters. For a more detailed description of the dynamics of the processes, we included in the text tables information about the absolute and relative changes of the average inflation and healthcare sector inflation. The average absolute growth of some parameters was calculated and discussed as a parameter describing the main trends of the processes.

The average absolute growth was calculated using the following formula:(y_n_ − y_0_)/(n − 1),
where
y_n_—last value of the parameter—in this case in 2022;y_0_—first value of the parameter in 2019;n—number of values (years = 4).


The ratio of the monthly wages needed for monthly therapy for originals compared to biosimilars was calculated and denoted as (O/B). The ratio of the percentage of pension needed for monthly therapy for originals compared to biosimilars was the same as the O/B ratio for employees.

All costs are presented in the national currency (BGN) at the exchange rate of EUR 1 = BGN 1.958.

## 3. Results

### 3.1. Economic Framework of Bulgaria during 2019–2022

The Bulgarian economy suffered from high inflation during the study period. The average NHIF budget for pharmaceuticals increased significantly, with an average of 42% less in 2019 in comparison with 2022. At the same time, inflation in the healthcare sector rose from 1.3% to 3.6%, and the national average inflation rate rose from 3.1% to 15.3%. Personal income increased by 34% (from BGN 1261 to 1693) and pension income by 103% (from BGN 348 to 709) (Table 1).

The dynamics of the explored parameters are presented in Table 2.

Rising inflation in the healthcare and pharmaceutical sectors, as well as average inflation in the country, might negatively influence affordability. This is especially true for people with chronic diseases who have to pay for long-term therapy. The NHIF reimburses most medicines, but there are still expensive medicines, and among those are biologicals for inflammatory diseases, which are not fully reimbursed. We have to note that general inflation was higher than that of healthcare and medicines, and the growth in income was higher than the inflation rate, which might be considered a relatively positive tendency for affordability.

### 3.2. Bulgarian Regulation of Managed Entry Agreements during 2019—2022

The NCPRMP approves prices and reimburses a share of prescribed medicines. The NHIF reimburses 25%, 50%, 75%, or 100% of the package price based on the reference price per the defined daily dose (DDD). The lowest price per DDD is accepted as reference and all medicines are reimbursed at that price. The rest is co-paid by the patients.

Legislative changes related to managed entry agreements are listed in Table 3. Payment institutions enters agreements with pharmaceutical companies to agree on discounts, company co-financing or budget caps, and payback rules to manage to lower prices and lower reimbursement costs. The price decrease should benefit both public spending and patients’ co-payments [19,20]. The extracted data mainly covers the regulatory mechanisms in force in the period of 2019–2022, which are formally published as regulations. 

Regulatory practice is quite diverse and oriented towards price and cost decreases, mainly in favor of the NHIF. The only exception is the companies’ co-payments for biological products, which directly affect patient access. The overall number of reimbursed biologics for inflammatory disease therapy was 12 branded and 4 biosimilar products during 2019–2022, 75% of which were paid by the NHIF and co-paid by the companies. The major regulatory changes address rebates and negotiations and are introduced as a mandatory requirement for pharmaceutical companies. 

### 3.3. Affordability of Medicines for Inflammatory Disease Therapy from Patients’ Point of View

Table 4 presents the average annual and monthly patient co-payment per dose.

Large variations in cost of therapy existed between the compared medicinal products and between the same medicinal products after the introduction of biosimilars. The most significant difference in patients’ co-payments was found in the treatment with the highest and the lowest price—infliximab. Adalimumab achieved the highest decline in cost of therapy between 2019 and 2022. We can also point out that in 2019 only two biosimilars were reimbursed on the Positive Drug List for inflammatory diseases, whereas in 2022 the overall number of biosimilars was 4.

Monthly income in Bulgaria increased in recent years, as did pensions. However, incomes remained much lower relative to the co-payment needed for therapy for patients with inflammatory diseases. The share of co-payments was examined in both groups—working-age patients and pensioners.

The monthly income of pensioners needed for therapy co-payment varied between 10% and 280%, which is impossible for patients to cover (Table 5). The hours of work to pay for a package also varied between 7 and 137 per month. The mean range of treatment with originators (infliximab, adalimumab, etanercept, and rituximab) for retired persons was 75.95 ± 15.13, whereas the mean range of treatment with originators for employees was 29.26 ± 9.25. The O/B ratio for 2022 was between 0.21 and 5.22 for employees or an average of 2.52 ± 2.11. The mean range of treatment went down during the study period, with decrease in the range of working hours needed for treatment (9 to 40.4 h), but it remained significantly high. The mean value of working hours needed to pay for chosen medicines was 19. 16 ± 13.10.7 out of all of the presented original medicines, exceeding the monthly income of pensioners. The decrease in the range of personal income needed for treatment was between 18.95% and 96.12%. The mean amount of pension needed to pay for the medicines was 59.11 ± 27.79%. For biosimilar rituximab, the monthly wages needed for a package and the percentage of pension needed for monthly therapy increased in the presented period by 10.5 h and 3.77%, respectively. That is why the companies agreed voluntarily to cover patients’ co-payments for therapy to ensure access and affordability.

## 4. Discussion

Countries apply different cost-containment measures to ensure better access to and affordability of therapy. In recent years, managed entry agreements have been constantly introduced that might be combined with other cost-containment measures. However, the impact of those measures on affordability needs to be studied [5,6].

In this study, we tried to shed light on the affordability of biological therapy within the framework of the general economic situation and regulatory changes aiming to improve patients’ access to and affordability of biologicals in Bulgaria. To the best of our knowledge, this is the only Bulgarian study examining simultaneously those three aspects of affordability improvement measures for patients. 

The study found economic declines with very high inflation, slowly increasing income for individuals, decreasing GDP, and growing public spending for pharmaceuticals. Average salary and pension rose during the period of 2019–2022 (0.34% for employees and 0.75% for retired persons) but not so significantly compared with the inflation of 15%. In Bulgaria, there is no mechanism implemented to protect patients from high charges, such as caps on co-payments or generic substitutions. At the same time, the VAT for pharmaceuticals is 20%—among the highest in the EU [21].

A variety of managed entry agreements have been introduced in Bulgaria, but the discounting policy prevails in almost all regulatory measures. Managed entry agreements affect mostly the industry pricing and reimbursement policy. An agreement with the industry to co-pay 25% for biological therapies is the only measure directly affecting the affordability for patients. The other measures affect the overall reimbursed cost of therapy. The OECD report revealed that in most EU countries, including Poland, Bulgaria, and Romania, financial- and performance-based managed entry agreements are available [22]. Slovakia recently adopted a new regulation in terms of pricing and reimbursement, and the advantages of the new introduced mechanisms are still being investigated [23].

The co-payment assistance programs for patients treated with biologicals lead to better treatment adherence and persistence, which have been assessed as potential reasons for better clinical outcomes [24]. The utilization of biologics in Poland, Bulgaria, and Romania is lower than in Hungary, Slovakia, and the Czech Republic. The variations in reimbursement and national policy are discussed as the main reason [25]. Recent data revealed that sales of biosimilars increased in 2020 in Slovakia, Hungary, and the Czech Republic and that they were higher than what was found for Poland [26].

Evidence suggests that MEAs are used in case the product poses a risk to the budget or the pricing and reimbursement rule creates a barrier to market entry [27,28]. There is no doubt that both reimbursement authorities and producers want to bring the products to the market, especially if their added value is undoubted. In case there is a lack of enough evidence of the product’s value, the reimbursement authorities try to either negotiate the prices, create patient access schemes, or apply other MEAs [29]. The reasons for using MEAs differ from the point of view of manufacturers and policymakers. For the manufacturer, faster access to the market ensures revenue, maintains innovation capacity, prolongs the market life of the products, and allows for high prices. Reimbursement authorities are concerned with patients with unsatisfied needs, ensuring access to the best possible therapy for the reimbursed budget, and, from the other side, containing costs for society. MEAs are crucial in ensuring faster patient access to innovative medicines with added value that satisfy unmet needs.

The main point explored in this study is the results of the introduction of MEAs for the improvement of patients’ access to biological therapy. It is the authors’ opinion that in case of biological therapy, combining different MEA approaches in the Bulgarian market is a successful policy.

Access to reimbursed biological disease-modifying antirheumatic drugs (bDMARDs) remains unequal across the European region. With the introduction of biosimilars access to therapy has increased, but nationally developed criteria play an important role in the establishment of eligible patients and healthcare costs [30]. Branded medicines for the treatment of cancer are less affordable than the lowest priced generic, according to a similar study in Pakistan. Only for the patients with high income are branded medicines affordable. The results also show that non-biological medicines were more affordable (58.1%) compared with biological medicines, especially for low-income patients [31].

A study in Slovakia found a 25–35% decrease in the price of biosimilars compared to originators, which lead to substantial cost savings in the amount of EUR 8–12 million for highly utilized products such as adalimumab [32]. The entry of the biosimilars of infliximab, adalimumab, and etanercept led to a weighted average price reduction per DDD of 13.6%, 0.9%, and 9.3%, respectively, and the entry of secondary biosimilars, reducing the price by 26.4%, 27.3%, and 9.1%, respectively. This was also accompanied by an increasing use of biosimilars of 88.9%, 14.6%, and 22.4% for infliximab, etanercept, and adalimumab, respectively [33]. In Bulgaria, there was also a decrease in the reference price per DDD for infliximab and adalimumab, as well as increased utilization of adalimumab after the introduction of a first biosimilar (from 0.25 to 0.29 DDD/1000 inhabitants per day) [34]. The current study’s results reveal that after biosimilar introduction the price of originators declined. We can also mention that the difference between the lowest priced medicines and originators was about 3 times, which is important for patients’ access and public spending on biologics. This is one more piece of evidence that, in the highly regulated pharmaceutical market, generic competition is one of the most effective market mechanisms to ensure price decreases.

Patients’ co-payments for treatment with biologics was compared under the assumption that patients pay for their own treatment, as is established in Annex 1, PDL. The results predicted an extremely high level of co-payment, which puts in question the affordability of therapy in Bulgaria [35]. Thus, the main burden falls on the pharmaceutical companies, which have to negotiate discounts, reduce prices, and cover the co-payments of patients when treated with biological products.

Real-world data revealed that biologics are effective for the treatment of RA, and non-TNFi and JAKi demonstrated the highest effectiveness [36]. Biologic disease-modifying antirheumatic drugs (bDMARDs) have been shown to be an effective treatment for rheumatoid arthritis, with better clinical and radiographical outcomes, but due to their high cost, they are only recommended for patients with an insufficient response to conventional therapy [37,38]. Patients’ quality of life, measured through the VAS and EQ5D scales, showed significant improvement despite the high incremental cost-effectiveness ratio and unfavorable cost-effectiveness when considering the treatment of RA [39].

High launch prices for biologics may impose confidential price discounts through managed entry agreements or suboptimal application of transparent or hidden access restrictions to ensure financial sustainability [40]. Countries apply different measures to improve patient access to medicines, but most of them are primarily aimed at containing the costs for payers. Among the main barriers to ensuring equal and faster access to therapy is the tendency of companies to market new products first in more wealthy markets and delaying access in other countries. Another obstacle is the willingness of the paying institution to control medicine costs and create artificial barriers to access [41].

## 5. Conclusions

The global economic crisis has affected Bulgaria and led to worsening economic parameters. Moreover, there are still no well-established practices to control public spending, as the measures taken to reduce the final cost of medicines mainly affect the pharmaceutical companies. It places a burden on manufacturers and creates a risk of product recalls. On the other hand, such cost-containment policies provide an opportunity for innovative treatment with biologicals for patients with inflammatory diseases. Most of the therapies cost more than patients’ monthly income.

## Figures and Tables

**Table 1 healthcare-11-02427-t001:** Economic indicators in Bulgaria during 2019–2022.

Indicator	2019	2020	2021	2022	Mean Value ± SD
Average inflation (%) *	3.1	1.7	3.3	15.3	5.85 ± 6.34
Healthcare sector inflation (%) *	1.3	2.8	0.8	3.2	2.03 ± 1.15
Inflation of medicinal product prices (%) *	1.3	3.2	0.7	3.2	2.1 ± 1.29
Inflation of doctors’ visits (%) *	1.8	1.7	4.9	5.5	3.47 ± 2.01
Average wage annual growth rate (%) *	10.6	9.7	12.3	12.8	11.35 ± 1.44
Change in GDP (%) *	4	−4	7.6	3.4	2.75 ± 4.86
Aaverage wage, BGN **	1261	1366	1530	1693	1462.5 ± 189.38
Average monthly pension (all types of pensions) (BGN)	383.03	437.17	536.71	674.54	507.86 ± 153
NHIF budget for pharmaceuticals, thousands (BGN)	4,299,603	4,744,705	5,084,871	6,116,305	5,207,954 ± 5,899,161
Reimbursed cost for rheumatology products paid for by NHIF (BGN)	84,713,447	91,927,096	95,328,275	102,450,770	93,604,897 ± 7,373,204

* Average annual inflation and key indicators are calculated each year based on data from the previous year; ** the average monthly salary is calculated as an average of annual data.

**Table 2 healthcare-11-02427-t002:** Dynamics of the average inflation and healthcare system inflation.

	2019 [%]	2020 [%]	2021 [%]	2022 [%]	
Dynamics of the Average Inflation and Healthcare Sector Inflation					
Average inflation	3.1	−1.4	0.2	12.2	According to 2019
Average inflation	3.1	−1.4	1.5	12	According to the previous year
Healthcare sector inflation	1.3	1.5	−0.5	1.9	According to 2019
Healthcare sector inflation	1.3	1.5	−2	2.4	According to the previous year
Dynamics of the Average Wage Annual Growth Rate					
Average wage annual growth rate	10.6	−0.9	1.7	2.2	According to 2019
Average wage annual growth rate	10.6	−0.9	2.6	0.5	According to the previous year

**Table 3 healthcare-11-02427-t003:** Managed entry agreements in Bulgaria in force during 2019–2022.

Introduced Managed Entry Agreements
A budget cap and a payback mechanism were introduced in case of budget excess. The reimbursed medicines are separated into three groups: -Group A—medicines for outpatient treatment prescribed after approval by a committee of three specialists; -Group B—all other medicines out of group A; -Group C—oncology and life-saving medicines. The maximum reimbursed budget for each group is negotiated with the marketing authorization holders four times annually. If the reimbursed budget in the group exceeds the negotiated cost, pharmaceutical companies return revenue respective to the proportion of their market share (implemented in 2018 and still in force).
The NHIF pays only for medicinal products for which companies provide discounts, established by Health Insurance Act (implemented in 2019 and still in force).
The NHIF conducts annual mandatory negotiation of discounts for medicinal products from the PDL. Discounts for single-entry products are negotiated and for generic products the reference reimbursement is applied at the lowest price per DDD for all medicinal products with the same INN (implemented in 2019 and still in force).
The NHIF negotiates discounts of no less than 10% for single-entry products with marketing authorization holders, entirely in favor of the NHIF. Discounts may be different for different therapeutic indications of the same medicinal product (implemented in 2019 and still in force).
The NHIF shall negotiate the discount on the cost per package calculated on the basis of the reference price of the respective medicinal product, paid for partially by the NHIF. In addition, for biological products for inflammatory diseases that are reimbursed at 75%, the companies voluntary agreed to co-pay the remaining 25% directly to the NHIF (implemented in 2019 and still in force).
For biological medicinal products for which a special diagnostic test is needed for differential diagnosis, the companies pay for the accompanying diagnostic to receive reimbursement status (implemented in 2019 and still in force).
Therapeutic effect monitoring is introduced and performance practice is paid for. The NHIF and the marketing authorization holder may discuss the reimbursed price based on the observed outcome of therapy. The contract includes specified clinical indicators in accordance with the criteria for monitoring set by the National Council on Prices and Reimbursement of Medicinal Products (implemented in 2020 and still in force).

**Table 4 healthcare-11-02427-t004:** An average patient’s co-payment per dosage per month during 2019–2022.

INN	2019				2020				2021				2022			
	NHIF Cost per Dose	Patient Co-Payment per Dose	Average Annual Patient Costs	Average Monthly Co-Payment	NHIF Cost per Dose	Patient Co-Payment per Dose	Average Annual Patient Costs	Average Monthly Co-Payment	NHIF Cost per Dose	Patient Co-Payment per Dose	Average Annual Patient Costs	Average Monthly Co-Payment	NHIF Cost per Dose	Patient Co-Payment per Dose	Average Annual Patient Costs	Average Monthly Co-Payment
Infliximbab biosimilar	358.42	119.47	3136.09	261.34	358.42	119.47	3136.09	261.34	279.27	93.09	2443.61	203.63	295.59	98.53	2586.40	215.53
Infliximbabbiologic	358.42	451.63	11,855.20	987.93	358.42	451.63	11,855.20	987.93	279.27	530.78	13,932.90	1161.10	295.55	514.50	13,506.00	1125.46
Adalimumab biosimilar	-	-	-	-	-	-	-	-	-	-	-	-	534.17	89.04	2314.90	192.91
Adalimumab biologic	789.02	263.01	6838.26	569.86	789.02	263.01	6838.26	569.86	697.02	355.01	9230.26	769.19	534.17	258.93	6732.20	561.02
Cetrolizumab	1137.92	189.66	5689.65	474.14	1072.03	178.68	5360.25	446.69	1072.03	178.68	5360.25	446.69	1070.85	178.48	5354.40	446.20
Secukinumab	1644.24	274.04	4384.64	365.39	1633.15	272.19	4355.04	362.92	1623.91	270.65	4330.40	360.87	1530.96	255.16	4082.60	340.21
Etanercept biosimilar	1151.63	95.97	4990.44	415.87	737.07	61.42	3193.97	266.16	737.07	61.42	3193.97	266.16	636.99	53.08	2760.30	230.02
Etanercept biologic	-	-	-	-	737.07	148.50	7721.87	643.49	737.07	148.50	7721.87	643.49	636.99	94.33	4904.90	408.74
Rituximab biosimilar	1170.19	1088.84	2177.68	181.47	1274.20	849.48	1698.96	141.58	1274.20	849.48	1698.96	141.58	663.81	2070.26	4140.50	345.04
Rituximab biologic	1170.19	1642.26	3284.52	273.71	-	-	-	-	-	-	-	-	663.81	442.54	885.08	73.76
Baricitinib	5671.58	1890.52	7562.08	630.17	5671.58	1890.52	7562.08	630.17	5671.58	1890.52	7562.08	630.17	5671.58	1890.52	7562.10	630.17
Tocilizumab	479.16	180.73	7048.47	587.37	473.30	186.59	7277.01	606.42	451.11	176.59	6887.01	573.92	431.91	168.46	6569.90	547.50
Golimumab	1298.97	432.99	1731.96	144.33	1205.00	401.67	1606.68	133.89	1157.00	385.67	1542.68	128.56	1136.80	378.94	1515.80	126.31
Ixekizumab	5111.93	567.99	10,223.82	851.99	5111.93	567.99	10,223.82	851.99	5111.93	567.99	10,223.80	851.99	5111.93	567.99	10,224.00	851.99
Ustekinumab	3790.33	1263.45	7580.70	631.73	3790.33	1263.45	7580.70	631.73	3552.13	1184.05	7104.30	592.03	3552.13	1184.05	7104.30	592.03
Tofacitinib	1111.77	370.60	4817.80	401.48	1111.77	370.60	4817.80	401.48	1043.41	438.96	5706.48	475.54	1010.70	417.62	5429.10	452.42

**Table 5 healthcare-11-02427-t005:** The monthly wages needed for a package and the percentage of pension needed for monthly therapy.

Medicinal Product	2019	2020	2021	2022		2019	2020	2021	2022		Range for Employees	Range for Retirees %	O/B Ratio for Employees
	% Of Personal Income (Retirement)	% Of Personal Income (Retirement)	% Of Personal Income (Retirement)	% Of Personal Income (Retirement)	Average Absolute Growth	Hours of Work for a Package (Hours)	Hours of Work for a Package (Hours)	Hours of Work for a Package (Hours)	Hours of Work for a Package (Hours)	Average Absolute Growth			
**Infliximab biosimilar lowest price**	68.23	59.78	37.94	31.95	−4.69	36.48	33.67	23.42	22.41	−12.09	14.07	36.28	-
**Infliximab originator**	*	*	*	*	−6.96	137.89	127.29	133.56	117.00	−30.36	20.89	91.08	5.22
**Adalimumab biosimilar lowest price**	-	-	-	28.60		-	-	-	20.05		-	-	-
**Adalimumab originator**	*	*	*	83.17	−7.07	79.54	73.42	88.48	58.32	−21.87	21.22	65.61	2.91
**Cetrolizumab**	*	*	83.23	66.15	−6.59	66.18	57.55	51.38	46.39	−19.21	19.79	57.64	-
**Secukinumab**	95.39	83.02	67.24	50.44	−5.21	51.00	46.76	41.51	35.37	−14.98	15.63	44.95	-
**Etanercept biosimilar lowest price**	*	60.88	49.59	34.10	−11.37	58.04	34.29	30.62	23.91	−24.82	34.13	74.47	-
**Etanercept originator**	-	*	*	60.60	−13.47	-	82.91	74.02	42.49	−28.86	40.42	86.59	1.77
**Rituximab biosimilar lowest price**	47.38	32.39	26.38	51.15	3.51	25.33	18.24	16.29	35.87	1.25	−10.54	−3.77	-
**Rituximab originator**	71.46	-	-	10.93	−10.17	38.20	-	-	7.67	−20.17	30.53	60.53	0.21
**Baricitinib**	*	*	*	93.42	−7.48	87.95	81.19	72.49	65.51	−23.7	22.44	71.10	
**Tocilizumab**	*	*	*	81.17	−8.35	81.98	78.13	66.02	56.92	−24.06	25.06	72.18	
**Golimumab**	37.68	30.63	23.95	18.73	−2.34	20.14	17.25	14.79	13.13	−6.31	7.01	18.95	
**Ixekizumab**	*	*	*	*	−10.11	118.91	109.77	98.01	88.57	−32.04	30.34	96.12	
**Ustekinumab**	*	*	*	87.77	−8.87	88.17	81.39	68.10	61.55	−25.72	26.62	77.16	
**Tofacitinib**	*	91.84	88.60	67.07	−3	56.04	51.73	54.70	47.03	−12.58	9.01	37.75	
**Mean value**					−6.811					−19.70	19.16	59.11	2.52
**SD**					4.162					9.26	13.10	27.79	2.11

* Exceeds monthly income.

## Data Availability

Data are available upon reasonable request from the authors.

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
