# Peer review of "How Managed Entry Agreements Influence the Patients’ Affordability to Biological Medicines—Bulgarian Example"

_healthcare, 2023, doi:10.3390/healthcare11172427_

Round 1

Reviewer 1 Report

The study encompasses both economic and regulatory analyses, focusing on the availability and accessibility of biological therapies in Bulgaria from 2019 to 2022. Using macroeconomic indicators and costs of reimbursed biological therapies, the accessibility for patients was estimated in terms of the number of hours of work required to finance monthly therapy. The increase in the National Health Insurance Fund (NHIF) budget for medications and inflation in the healthcare sector significantly impacted their accessibility. The presence of co-payments may result in a substantial commitment of patients' income towards purchasing medicines. Often, retirees require from 10% to 280% of their monthly income to cover the co-payment for therapy.

The global economic crisis has negatively affected Bulgaria's economic indicators. However, well-established practices for controlling public spending are lacking. The costs of most therapies exceed patients' monthly incomes. Different countries employ diverse cost-control measures to ensure better access to and affordability of therapy. In recent years, numerous managed entry agreements have been introduced to manage drug market entry, which can be combined with other cost-control measures. Nonetheless, the impact of these actions on medication accessibility requires further investigation.

This study concentrates on the availability of biological therapy within the context of the overall economic situation and regulatory changes aimed at enhancing access and affordability of biological therapies in Bulgaria. It is the only Bulgarian research work that simultaneously analyzes these three aspects of improving patient accessibility.

The study results demonstrate an economic decline alongside very high inflation, with individual incomes slowly increasing, contrasting with a decrease in GDP and a rise in public spending on medications. The introduction of biosimilars has increased therapy accessibility, yet national criteria play a vital role in determining eligible patients and healthcare costs.

The policy of price reduction predominates in managed entry agreements. These agreements primarily influence the pricing and reimbursement policies of the pharmaceutical industry. Introducing a 25% co-payment for biological therapies is the only measure directly affecting patient accessibility. Other measures impact the overall reimbursed cost of therapy.

In conclusion, the study highlights that despite the implementation of various cost-control measures, accessibility to biological therapy remains a challenge in Bulgaria. The impact of the global economic crisis, lack of effective practices for controlling public spending, and high medication costs are significant issues requiring further action to improve the patient situation.

(Personal observations by the reviewer) The study constitutes a significant contribution to understanding the issue of the availability and accessibility of biological therapies in Bulgaria. The presented results shed light on the difficulties faced by patients and underscore the necessity for further regulatory and economic actions to ensure equitable access to effective therapies.

Suggestions and comments:

1.       Table 1 - A relevant comparison seems to be the statistical comparison, e.g., The average inflation (%) vs. Healthcare sector inflation (%) and An average wage, BGN vs. Average monthly pension (all types of pensions) (BGN), e.g., using a structure test for the percentage value.

2.       Table 2 - It is challenging to read - the data presented should be minimized to the most essential points.

3.       Table 4 - Statistical analysis (e.g., using a % structure test) should be performed and relevant dependencies or their absence in the table should be appropriately marked. The statistical significance level (e.g., p < 0.05) is missing.

4.       Why does the References section switch to Roman numerals (XXI and XXII) after entry 20?

5. The Discussion section lacks references to neighbouring countries with a similar market structure, such as Poland, Slovakia, and the Czech Republic.

Author Response

Dear reviewer,

thank you very much for the recommendations!

Kind regards,

Zornitsa

Reviewer 2 Report

I believe, Managed Entry Agreements (MEA) are an unfortunate attempt to emulate a market mechanism restricted by a goal (equitable distribution of experimental medications) imposed by purely ideological reasons. This ideology (socialism) is proven in numerous countries' experiments to be a failure. So, I am deeply convinced, this experiment is useless at best.

Irrespective of my opinion the experiment is already in progress in the European Union (EU) countries.  Under these circumstances, the right thing to do for responsible economists is to provide as many detailed countries’ protocols of current experiments as possible.

And the authors of the reviewed manuscript have just done one of the country case studies.

So, the issue of the article is highly relevant, data collected by authors are undoubtedly very useful and deserves to be shared for further discussion.

The study's methodology is relevant to the challenge and I am not sure it is possible to improve it significantly in a reasonable time with limited resources. Instead of an extended and detailed analysis of the “advantages” and "disadvantages" of various MEA instruments authors provided a statistical analysis of observable outcomes of MEA application in Bulgaria. The outcomes are measured as prices of selected medications weighed on pensioners’ average monthly income.

I am convinced, that the research program of studies of MEA will benefit greatly from comparing the MEA mechanism to previously existing in developed countries' market solutions.

Real market solutions include strong interests of health insurers to avoid responsibility for the obviously irresponsible (too risky) choices of the patient and his physician. Insurers, practitioners, and even patients rely on private expertise on the medications' quality instead of FDA-type bureaucratic power. The overall efficiency of the latter is challenged by empirical studies since at least Peltzman, 1987.

So generally, rich, and more or less “risk-loving” people are naturally volunteered for testing of new medications. After medication proved its efficiency interest to earn more incentivise to produce many more and sell cheaper in order to mitigate risks of competition. After patent protection is over (usually after 20 years) competition quickly undercuts the prices to quite an affordable level.  Before it, charities provide highly focused support for the neediest families.

So, the question is – does MEA succeed to provide better access for patients to medications than a free-market solution or not?

I would not insist on the immediate inclusion of this ‘base for comparison’ in the reviewed paper, but I would strongly recommend the authors at least mention such a procedure as a component of their future studies and as the current study's limitation.

I guess, placing some figures presenting the principal outcomes on the affordability of medications would be helpful.

 Peltzman, S. 1987. The Health Effects of Mandatory Prescriptions. The Journal of Law & Economics, Vol. 30, No. 2 (Oct., 1987), pp. 207-238

Author Response

Dear reviewer,

thank you very much for the recommendations!

I was really pleased to read your comments.

1) I believe, Managed Entry Agreements (MEA) are an unfortunate attempt to emulate a market mechanism restricted by a goal (equitable distribution of experimental medications) imposed by purely ideological reasons. This ideology (socialism) is proven in numerous countries' experiments to be a failure. So, I am deeply convinced, this experiment is useless at best.

Irrespective of my opinion the experiment is already in progress in the European Union (EU) countries.  Under these circumstances, the right thing to do for responsible economists is to provide as many detailed countries’ protocols of current experiments as possible.

And the authors of the reviewed manuscript have just done one of the country case studies.

So, the issue of the article is highly relevant, data collected by authors are undoubtedly very useful and deserves to be shared for further discussion.

Dear reviewer, thank you for your comment. We for sure can have an ideological discussion for the basis of MEAs measures, because during the socialism there was no medicines, so there was no risk to share between the companies and the public budget. We can even go further that the MEAs were introduced in wealthiest economic countries as a partnership between the government and industry. These are the impressions from authors like Panos Kanavos from London School of Economics and Alexandra Ferrario his former PhD student, cited in this work. This discussion goes beyond the scope of our work, but you gave us a brilliant idea to explore the social and even political background of the introduction of MEAs in different countries.  

2) The study's methodology is relevant to the challenge and I am not sure it is possible to improve it significantly in a reasonable time with limited resources. Instead of an extended and detailed analysis of the “advantages” and "disadvantages" of various MEA instruments authors provided a statistical analysis of observable outcomes of MEA application in Bulgaria. The outcomes are measured as prices of selected medications weighed on pensioners’ average monthly income.

I am convinced, that the research program of studies of MEA will benefit greatly from comparing the MEA mechanism to previously existing in developed countries' market solutions.

The following comment was added in the discussion section.

Evidence suggests that MEAs are used in case the product is posing a risk for the budget or pricing and reimbursement rule creates barrier in front its market entry [27], [28]. There is no doubt that both reimbursement authorities and producers want to bring the products to the market, especially if their added value is undoubtful. In case if there is lack of enough evidence for the product value the reimbursement authorities are trying to either negotiate the prices, create patient access schemes or apply other MEAs. [29]. The reasons to use MEAs differs from the point of view of manufacturers and policy makers. For the manufacturer faster access to the market ensures revenue, maintain innovation capacity, prolong market life of the products, allows high prices. Reimbursement authorities are concerned by the patients with unsatisfied need, en-suring access to the best possible therapy for the reimbursed budget, and from the other site contain cost for the society. MEAs are crucial in ensuring faster patients access to innovative medicines with added value, that satisfy unmet needs.

3) Real market solutions include strong interests of health insurers to avoid responsibility for the obviously irresponsible (too risky) choices of the patient and his physician. Insurers, practitioners, and even patients rely on private expertise on the medications' quality instead of FDA-type bureaucratic power. The overall efficiency of the latter is challenged by empirical studies since at least Peltzman, 1987.

Fully agree, but in Europe the pharmaceutical market is highly regulated, and it is accepted by many authors that it is not the real market because the patient in its role of consumer does not directly meet the producer.

4) So generally, rich, and more or less “risk-loving” people are naturally volunteered for testing of new medications. After medication proved its efficiency interest to earn more incentivise to produce many more and sell cheaper in order to mitigate risks of competition. After patent protection is over (usually after 20 years) competition quickly undercuts the prices to quite an affordable level.  Before it, charities provide highly focused support for the neediest families.

Agree for the first part of the comment but in Europe the role of donator is taken by the government with its compulsory health insurance, and charity does not play a significant role in supporting peoples accessto medicines. The role of generic competition is highly recognised by many authors as one of the most effective measures in ensuring affordable medicines, but in case for patented medicines governments have to negotiate with the manufacturers.

We add such comment in the discussion section.

This is one more piece of evidence that in the highly regulated pharmaceutical market generic competition is one of the most effective market mechanisms to ensure price decrease.

5) So, the question is – does MEA succeed to provide better access for patients to medications than a free-market solution or not?

I would not insist on the immediate inclusion of this ‘base for comparison’ in the reviewed paper, but I would strongly recommend the authors at least mention such a procedure as a component of their future studies and as the current study's limitation.

Agree added in the discussion.

This the main point explored in this study, and it is our opinion that in case of biological therapy combining different MEAs approaches on the Bulgarian market is a successful policy.

6) I guess, placing some figures presenting the principal outcomes on the affordability of medications would be helpful.

 Peltzman, S. 1987. The Health Effects of Mandatory Prescriptions. The Journal of Law & Economics, Vol. 30, No. 2 (Oct., 1987), pp. 207-238

            We appreciate your proposal but this study is focusing on European type of prescribing practice that is regulated and controlled and does not afford patients influence on the physician.

Kind regards,

Zornitsa

Reviewer 3 Report

Thank you for the opportunity to review an article on state subsidies for patients to purchase biologic treatments to treat inflammatory diseases. Treatment with biologics significantly affects health, improves function. Expenditures both by the State and directly from household budgets are an important part of existence. The authors have reliably shown the systems of subsidies and reimbursement in an economic way, which is an important element in planning the next steps to enable reimbursement of treatment with new drugs.

My suggestions are:

1. Additions to the extent to which co-payments improve clinical outcomes for patients treated with biologic drugs.

2. To what extent subsidies affect patients' quality of life

3. On the technical side, there is no numbering of the literature from point 20 up wards

Author Response

(The authors gave the same response as above.)

Round 2

Reviewer 1 Report

The authors of the publication adequately addressed the provided comments.